# Resilience Mechanisms and Coping Strategies for Forcibly Displaced Youth: An Exploratory Rapid Review

**DOI:** 10.3390/ijerph21101347

**Published:** 2024-10-11

**Authors:** Akm Alamgir, Christopher Kyriakides, Andrew Johnson, Gemechu Abeshu, Bay Bahri, Miles Abssy

**Affiliations:** 1Access Alliance Multicultural Health and Community Services, Toronto, ON M5T 3A9, Canada; 2Department of Sociology, York University, Toronto, ON M3J 1P3, Canada; 3The Centre for Addiction and Mental Health, Toronto, ON M6J 1H4, Canada; 4University of Toronto, Toronto, ON M5S 1A1, Canada

**Keywords:** coping mechanism, ecology of resilience, refugee well-being, youth mental health, acculturation

## Abstract

**Context:** The global escalation of conflict, violence, and human rights violations sets a pressing backdrop for examining the resilience of forcibly displaced youth (FDY) in Canada. This study aims to unpack the multifaceted challenges and resilience mechanisms of FDY, focusing on their health, well-being, and integration into host communities. It seeks to identify current models of resilience, understand the factors within each model, and highlight gaps and limitations. **Methodology:** Using a university librarian-supported structured search strategy, this exploratory rapid review searched literature from Ovid Medline and open-source databases, published in English between January 2019 and January 2024, that fit specific inclusion criteria. The eligible articles (N = 12 out of 4096) were charted and analyzed by two student researchers with the Principal Investigator (PI). Charted data were analyzed thematically. **Results:** The selected studies captured diverse geographical perspectives, resilience models (such as Ungar’s ecological perspective and Masten’s resilience developmental models), as well as protective and promotive frameworks. Key findings indicate the complexity of resilience influenced by individual, familial, societal, and cultural factors. Each model offers insights into the dynamic interplay of these influences on FDY’s resilience. However, these models often fall short of addressing the nuances of cultural specificity, the impact of trauma, and the intersectionality of FDY’s identities. **Conclusions:** Recognizing the diverse and evolving nature of FDY’s coping mechanisms, this study advocates for a culturally appropriate approach to resilience that integrates an intersectionality framework of individual attributes and culturally sensitive models.

## 1. Introduction

In recent years, the world has witnessed an increased number of forcibly displaced individuals. By the end of 2023, more than 117.3 million people were forcibly displaced due to factors like persecution, violence, and human rights violations, marking the largest single-year increase in UNHCR’s history [1]. Conflicts such as the wars in Sudan, Syria, the Democratic Republic of Congo, Afghanistan, and Ukraine, as well as climate-driven displacements, have interrupted many integral components of these lives including the loss of a home, community, and a sense of normalcy [1]. Canada, a country recognized for its humanitarian stance and refugee resettlement programs, has faced a significant influx of forcibly displaced individuals. The nation has welcomed over a million refugees since 1980, with nearly 60,000 individuals in 2016 [2]. The average age of a refugee in 2016 was 28.9 years old [2], which indicates that a significant number of young individuals come to Canada. This study explores refugee youth’s health, well-being, and integration challenges.

In Canada, different jurisdictions build their operational definitions of ‘youth’ regarding age group [3,4,5,6,7,8]. Considering all of these definitions, we define, in this study, ‘youth’ as a person aged 15 to 24 years, following the lead of Canadian and international institutions. From a different perspective, Jones [9] (p. 11) refers to ‘youth’ as a social construction characterized by transitional functional, constitutional, emotional, and behavioural attributes from children to adults, and argues framing of youth by age range.

Youth are especially affected by displacement and present unique challenges. Forcibly displaced youths (FDYs) face restricted access to education and livelihoods, and many assume critical roles within their families, such as caretakers or ‘breadwinners’ [9,10]. FDYs “… have been forced to flee their homes because of armed conflict, generalized violence, violations of human rights or natural or human-made disasters, and who have not crossed an internationally recognized state border” [11]. FDYs also tend to confront risks specific to their gender, with men potentially facing recruitment into armed groups, or women at risk of early sex or survival sex practices [8].

Despite challenges, FDYs demonstrate remarkable resilience. Resilience in this context refers to the capacity to withstand, adapt to, and recover from the significant adversities they face [12]. This resilience is not about enduring hardship but involves actively engaging with challenges to mitigate their effects and foster positive outcomes [13]. This engagement involves a process where FDYs leverage internal and external resources to navigate their circumstances effectively.

FDYs often develop a unique set of skills and coping mechanisms. These include the ability to quickly adapt to new roles and environments, finding innovative solutions to complex problems, and maintaining hope and optimism [14,15]. Moreover, the concept of resilience in FDYs extends beyond individual capacities, into their communities and the broader socio-political context [16]. This perspective recognizes that resilience is not just an individual trait but a result of interactions between individuals and their environments. Therefore, understanding resilience in FDYs requires a holistic approach that considers both personal strengths and external factors that can support their resilience.

This paper uses existing literature to identify established theories and models of resilience, the contributing factors that influence resilience, and effective strategies used to build resilience. This study seeks to answer the following research questions: What are the current models of resilience regarding refugee youth? What are the factors included in each model? What are the gaps and limitations in each model?

## 2. Methods

The Immigration and Refugee Board of Canada shows that annual refugee claimant arrivals in Canada have seen a substantial rise over the last decade increasing tenfold from 29,825 in 2013 to 137,529 by 2023 [17], not including Ukrainian refugees who came with temporary resident status. Such an increase, which structurally created a crisis in the availability of shelter beds, led to many newly arrived refugee claimants sleeping outdoors, as well as several high-profile cases involving the tragic death of individuals seeking access to shelter services. The refugees and researchers working with refugees use different terms to report their unique challenges. The challenges faced by the refugee claimants led some scholars to question the emphasis placed on refugees’ resilience to hardship, due to which they were left to their own means to survive [18]. With this in mind, a team of researchers conducted an exploratory rapid review to appraise existing literature (published in English) on coping strategies and resilience among FDY populations.

A rapid review (RR) is an exploratory approach to a review of the literature that speeds up the analysis of published data [19]. Such a review option helped us understand the popular consensus, more popular terms, and context in the field. In other words, a distinguishing feature of a rapid review is the ability to conduct reviews within a limited timeframe. Scholars argue that a rapid review is preferable when the research aims to find an answer that will affect policy decisions or other pressing matters that are normally produced in a systematic review but have neither the time nor budget to complete a full systematic review [20,21].

Following a rapid review protocol, articles from all geographical locations were considered. The study team considered the sensitivity and specificity of the diverse refugee population in Canada and decided to set 15–24 years as the age of reference for defining youth. Studies were included if 80% of the study population was between the ages of 15–24 and were based on a conceptual framework or model of FDY resilience. The population of focus for this review comprised FDY in the post-migration phase. Excluded studies comprised non-forced migration, focused on pre- (or peri-) migration factors, involved a study population outside of the age criteria, or lacked relevance to FDY resilience. To ensure a contemporary understanding of the issues faced by these youth, the review was confined to studies conducted and published between January 2019 and January 2024.

We consulted OVID Medline and open-access databases. A structured search strategy was constructed with the assistance of a librarian from the University of Toronto. The Medline Search String, Boolean Operators, and Wild Cards are mentioned as Appendix A. The OVID Medline database was searched on 9 January 2024, with a date filter set from 2019 to current. Through recommendations from the librarian, the team was trained in developing the search strategy (strings, MeSH terms, Boolean Operators, wild card, etc.) using OVID Medline and open-access resources. The search string was broken down into six categories that included relevant synonyms for search content (psychological adaptation, resilience or coping strategy, abandoned child, adolescent, immigrants or refugees, and asylum). The team also searched for any open-source peer-reviewed literature published during that time frame.

The initial search string resulted in 4096 results, which represented a wide spectrum of insights into the resilience of adaptation of FDY (Figure 1). Articles then went through a rigorous screening process and included articles (N = 12) that were then charted using a Microsoft Excel spreadsheet. This approach allowed for an organized comparison and contrast of the various models of resilience identified in the literature. The two field researchers (BB and MA), supervised by the Principal Investigator (AA), screened and charted selected articles for the next steps. The decision was made with consensus between the two field researchers. The PI was available to resolve differences of opinion between the field researchers. The review compiled emerging themes, which were carefully categorized and analyzed based on factors contributing to resilience. These were further classified into thematic clusters that emerged from the analysis. The charted models were grouped in matching thematic clusters.

## 3. Results

### 3.1. Geographic Make-Up of Studies

Studies reviewed in this article cover diverse nationalities, geographic boundaries, and cultural settings in their discussions about the theoretical frameworks used for understanding refugee youth resilience. For example, Dangmann et al. focused on Syrian refugees in Norway, while Garbade et al. assessed the resilience strategies and challenges faced by refugee youth in Germany [22,23]. The mentoring of unaccompanied migrant youth in the Barcelona Metropolitan Area was explored, providing insights into interventions in a European urban setting [24].

In Africa, Scharpf et al. studied the risk, protective, and promotive factors for the mental health of Burundian refugee children living in refugee camps for an in-depth look at the experiences of refugee children in camp settings [25].

Bennouna et al., in North America, explored the resilience of Middle Eastern and North African immigrant-origin youth in the U.S., focusing on acculturation and mental health in war-affected youth [26]. A snowball search with bibliographies found a study by Roesch et al., which examined coping strategies among Canadian secondary school students from immigrant backgrounds [14]. Khan et al. investigated factors impacting the mental health and resilience of homeless refugee youth in Toronto, Canada, while Gyan et al. analyzed resilience among immigrant and refugee youth in Montreal [27,28].

In Australia, Miller et al. studied the resilience, psychological distress, and coping strategies of high school students from refugee backgrounds and young adult Australian Hazaras [29]. These studies, along with research on the mental health care experiences of unaccompanied young refugees, offer insights into resilience-building strategies among forcibly displaced youth in Australia, providing a broad view of resilience across different contexts.

### 3.2. Current Models of Resilience

Several resilience models were identified during the search, such as four studies referenced Michael Ungar’s ecological perspective of resilience [22,26,28,29], three studies referenced protective models of resilience [22,25,30], two studies referenced promotive models of resilience [22,31], two studies referenced Masten’s Model of Resilience [31,32], two studies referenced Suárez-Orozco et al.’s Integrative Risk and Resilience Model [31,32], one study referenced Schwartz’s Rethinking of Acculturation Concept [23], one study referenced Phenomenological Variant of Ecological Systems Theory [32], one study referenced Silove’s ADAPT model [27], and one study referenced Litman’s 15 Scales of the COPE Inventory [33].

#### 3.2.1. Bronfenbrenner’s Ecological Systems Theory

This framework (Figure 2) argues that the development of a child is influenced by nested social systems. Moving from the smallest to largest scale, the individual, micro, meso, exo, and macro factors a child is surrounded with determine their developmental trajectories [34]. Many of the models of resilience are found in Bronfenbrenner’s Ecological Systems Theory, including Ungar’s ecological perspective, Masten’s developmental model, PVEST, and Suárez-Orozco et al.’s Integrative Risk and Resilience Model, along with protective and promotive resilience models [35]. These models were found in several research publications even though Bronfenbrenner’s theory itself might not have been explicitly mentioned in the search results [22,24,25,26,28,29,30,31]. The Ecological Systems Theory states that the development and behaviour of an individual is influenced by a series of interconnected factors, of varying levels of scope [34]. Bronfenbrenner uses categories such as the microsystem, mesosystem, exosystem, macrosystem, and chronosystem to stratify the scale of impact of such factors [34]. The micro-level describes all factors that impact youth on a personal or individual level, the meso-level includes factors that connect the individual to the community on a slightly larger scale, the exo-level includes factors that indirectly influence the individual on a societal scale, the macro-level includes socio-cultural factors which set societal standards, and the chronosystem comprises the transitions or shifts which occur over the lifetime of a child [34].

#### 3.2.2. Ungar’s Ecological Perspective of Resilience

Ungar’s ecological perspective of resilience is a conceptual framework connecting four central principles of resilience building to ecological systems: decentralism, complexity, atypicality, and cultural relativity [36]. Ungar calls for the decentralization of resilience, as he identified a major oversight of existing resilience definitions encompassing both the processes and outcomes [36]. Ungar argues that researchers must examine both the effects on the individual and the protective mechanisms that influence risk factors, to reduce their impact on the individual. The second principle, complexity, states that the protective processes that cause growth under adverse conditions are too complex to predict singular developmental outcomes [36]. Ungar asserts that resilience-promoting processes only seem to produce predictable outcomes when the likelihood of positive outcomes depends on the degree of risk imposed by the environment. The principle of atypicality states that protective processes contributing to resilience do not have binary outcomes (i.e., this behaviour is positive, and another is negative) as these concepts are context-specific [36]. Based on Bronfenbrenner’s Ecological Systems Theory, the principle of atypicality suggests that micro (individual), meso (community), exo (government and media), and macro (socio-cultural) factors can all have a significant influence on refugee youth resilience [36]. Lastly, cultural relativity accounts for differences in expression and experiences of resilience across cultures. Ungar acknowledges that resilience is unique and complex, suggesting that resilience may differ across cultures where access to different environmental resources differs [36].

#### 3.2.3. Masten’s Resilience Developmental Model

Grounded in Bronfenbrenner’s Developmental System Theory, Masten’s resilience developmental model aims to infer resilience based on two criteria: first, the individual must have experienced adversity that disrupts adaptive functioning and second the individual must adapt well, despite the negative life event [31,34]. Where the first criterion may seem straightforward, as forcibly displaced refugee minors will have experienced adverse life events, the second criterion is further divided into developmental and acculturative tasks, to establish a baseline to determine if individuals are adapting “well.”

Developmental tasks are expectations or standards for achievement and behaviour, set by family, teachers, and society [31]. These standards are developed around the normative principle which associates positive adaptation to what is “normal” for individuals of a certain age, gender, circumstance, and culture [31]. These developmental tasks can further be categorized as individual development, relationships with authority, relationships with peers, and functioning on a proximal and societal level [31]. Completing developmental tasks such as the development of self-regulation, having close friends, and finding success in school are examples of milestones that indicate adequate adaptation [31].

Furthermore, Masten’s model calls attention to acculturative tasks as immigrant and refugee youth must develop in their host culture and culture of origin [31]. Acculturative tasks such as having relationships, cultural understanding and nuance, and developing a strong identity in both host and native cultures are benchmark events for determining positive adaptation [31].

#### 3.2.4. Phenomenological Variant of Ecological Systems Theory (PVEST)

The Phenomenological Variant of Ecological Systems Theory (PVEST) proposed by Spencer et al. is a framework that integrates the experience of individuals and associates these experiences with self-organizational perspectives. Spencer et al. assert that the processing of phenomena not only influences self-esteem but also influences how individuals assign meaning and significance to different personal attributes such as physical attributes and behaviours [37]. The authors further suggest that perceptions of experiences in culture influence how an individual perceives themself [37].

Based on Bronfenbrenner’s ecological systems theory, the five core concepts of PVEST are (i) risk contributors, (ii) stress engagement, (iii) reactive coping mechanisms, (iv) stable coping responses, and (v) life stage outcomes (coping products, shape the way individuals perceive themselves and respond to stressful events) [34,37]. The authors describe risk contributors as the self-appraisal process individuals undergo in response to perceived stereotypes and biases. Factors such as race, sex, socio-economic status, biological characteristics, and physical status of individuals are risk factors. These qualities determine the types of prejudice a person may face based on overarching socio-cultural opinions [37]. The second concept, stress engagement, occurs when individuals face intermediate stressful events based on their physical and behavioural characteristics. These events are related to the availability of social support or the experience of daily hassles and/or community dangers [37]. The third factor, reactive coping mechanisms, identifies corrective problem-solving strategies used when individuals encounter a stressful event. Spencer et al. state that such strategies are either adaptive or maladaptive based on their impact on self-image and personal identity [37]. The fourth factor, stable coping responses, includes emergent identities and examines the identities emerging from reactive coping mechanisms. The integration of cultural goals and available means helps shape these coping responses, further identified as cultural identity, sex-role identity, self-efficacy, and personal identity [37]. These identities then help shape the resulting life-stage outcomes. The fifth factor, life stage outcomes, meaning the coping products, assesses the outcomes of perceived social and integrative stresses and categorizes individuals as either productively or adversely adjusted [37]. Productive individuals have general life satisfaction and good health, are competent in forming relationships, and exhibit effective parenting, whereas adverse individuals may have poorer health, mental illness, a lack of intimacy, and deviance [37].

#### 3.2.5. Integrative Risk and Resilience Model (Bicultural Socialization)

The Integrative Risk and Resilience Model proposed by Suárez-Orozco et al. suggests a nested web of concentric factors, which not only influence the resilience of migrants and refugees but also identify universal and immigrant-origin-specific tasks that determine positive adjustment and resilience [35].

The authors suggest that global forces such as war, violence, or environmental catastrophes act as catalysts for migration, and migrants including refugees are impacted by the political and social contexts of reception in host countries [35]. Factors of this category include attitudes toward migrants, refugee and asylum seeker resettlement services, and national policies targeting immigrants [32,35]. These overarching political and social contexts then influence microsystems such as neighbourhoods, schools, and family dynamics in FDY communities [32,35]. Lastly, the relationships generated in each respective microsystem influence the individual level where developmental competencies, intersecting social identity and positionality, and acculturative adaptation are regulated [35]. Individual tasks such as self-regulation, social relationships, secure sense of identity, psychological well-being, life satisfaction, and mental health are considered universal benchmarks to evaluate successful adaptation and resilience [35]. Immigrant-origin-specific tasks such as host country language acquisition, development of a sense of belonging, and acquisition of host culture norms are benchmarks designed to target refugee and immigrant youth adaptation and resilience [32,35].

#### 3.2.6. Schwartz’s Reconceptualization of Acculturation

Schwartz’s reconceptualization of acculturation seeks to understand the interplay of how migrants and refugees adopt certain aspects of the host country’s culture and maintain aspects of their own native culture. Schwart’s model expands on Berry’s model of acculturation, which casts receiving-culture acquisition and heritage-culture retention as dimensions that intersect into four acculturation categories: assimilation, separation, integration, and marginalization [38]. However, Schwartz argues that Berry’s model should be adjusted for variations among migrants and their circumstances and should represent a more nuanced approach [39]. Schwartz, like Berry, identifies four distinct acculturation strategies. Assimilation strategies occur when migrants and refugees leave behind native cultural values and adopt the traditions of their host countries [39].

In contrast, separation strategies occur when newcomers reject the host country’s culture and seek to solely maintain their original cultural identity [39]. In comparison to fully accepting and fully rejecting host culture, language, and norms, integration or biculturalism occurs when migrants and refugees seek to foster a cultural identity with both their host and native cultures [39]. Lastly, marginalization strategies occur when migrants and refugees are not involved in activities in either the host country or country of origin [39].

Schwartz’s model, however, further calls into question how individual-level, stress-related, and contextual factors influence migrant orientation toward the culture of either the host country or country of origin [23]. Schwartz describes individual-level factors as age and gender, stress-related factors as traumatic experiences or daily stressors, and contextual factors as social or familial support [39].

#### 3.2.7. Litman’s COPE Inventory

The COPE Inventory proposed by Litman aims to categorize coping strategies used by refugee youth into avoidance-oriented, emotion-focused, problem-focused, and socially supported [40]. The model describes avoidance strategies as actions youth take to avoid stressful events such as denial, behavioural disengagement, substance use, and self-distraction [40]. The next category describes emotionally related coping strategies such as positive reinterpretation, acceptance, humour, religiosity, and self-blame [40]. Litman describes problem-focused strategies as goal-oriented implementations, such as goal setting, planning, or active coping [40]. Lastly, the model proposes socially supported strategies such as expressing negative emotions, seeking comfort, and looking externally for help or support [40].

#### 3.2.8. Promotive and Protective Frameworks of Resilience

Referenced by Dangmann et al. and Motti-Stefanidi et al., promotive resilience mechanisms directly improve healthy functioning irrespective of risk exposure [22,31]. Also known as additive models, factors that exhibit promotive effects on FDY resilience and mental health are believed to directly add to the outcome of resilience, without interacting with risk factors FDY may face [31]. Promotive factors are commonly operationalized in research as factors that lead to increased adjustment or enhance the ability of FDY to “bounce back” when youth experience adverse events [31].

Unlike promotive factors, protective resilience mechanisms propose that certain factors directly mitigate the risk factors FDY may face. Further elaborated by Scharpf et al. and Scharpf et al., protective factors include the protective resources FDY can draw from, and these factors can be categorized by temporality and perceived impact on the likelihood of risk occurring [25,30].

#### 3.2.9. Silove’s ADAPT Model

The ADAPT model proposed by Silove is a framework for mental health and psychosocial engagement in post-forcibly displaced migratory settings. The framework aims to provide relevant psychosocial interventions to improve the mental health and resilience of beneficiaries [41]. The model is grounded by five core pillars: safety and security, bonds and networks, justice, roles, and identities, and existential meaning [41]. Silove argues that mass conflict underpins these psychosocial pillars of society. To help forcibly displaced individuals return to a standard level of functioning, each of these pillars must be satisfied through sensitized programming [41].

## 4. Discussion

### 4.1. Micro-Level Factors Contributing to Resilience

Micro-level factors contributing to resilience are operationalized on a large scale and account for the biological resources, cognitive resources, social-emotional resources, and temperament of FDY. These factors intersect with the individual characteristics of FDYs such as gender, age, ethnicity, legal status, socioeconomic status, and religion. In studies that were based on Ungar’s ecological perspective of resilience, protective resilience models and promotive resilience models operationalized factors on the individual level. These factors included positive self-perception, identity affirmation, agency, psychological well-being, hope, self-esteem, engagement with coping strategies, emotional regulation, support seeking, prosocial behaviour, independence, depression and anxiety symptoms, and psychological distress [22,25,27,29,30,32,35].

Characterized as developmental tasks by Masten’s model and Suárez-Orozco et al. [32], micro-level factors such as self-regulation, secure sense of identity, mental well-being, life satisfaction, self-esteem, and engagement with coping strategies were identified as checkpoints indicative of adequate adjustment and resilience [31,32,35].

Analyzed by Tardif-Grenier et al. [33], the authors conceptualized coping behaviour and resilience through the analysis of different actions. Researchers operationalized avoidance-oriented coping strategies as self-reported denial, disengagement, self-distraction, and substance use. The emotion-focused coping strategies are reported as self-reported acceptance, humour, religion, and self-blame; while the problem-focused coping strategies are self-reported active planning; and socially supported coping strategies as support and venting [33].

### 4.2. Meso-Level Factors

Meso-level factors are identified as factors that dictate the intermediate-level structures that exceed the individual but are not as expansive as society. Although each model and framework identify specific factors on the meso-level, topics surrounding family, education, and connection to the community were commonly identified.

### 4.3. Family

The role of family on resilience was consistently explored with many studies citing familial factors as both providing protective and risk-oriented qualities to FDY resiliency building. Studies that used Ungar’s Ecological Perspective of Resilience, protective resilience models, and promotive resilience models operationalized familial factors as familial support (protective), warm parent–child relationships (protective), negative parental behaviours (risk), harsh parenting (risk), family cohesion (protective), living with at least one biological parent (protective), parental abuse (risk), and family congruence [22,25,26,27,29,30,32].

Characterized as developmental tasks by studies that used Masten’s Resilience Developmental Model and the Suárez-Orozco et al. [35], Integrative Risk and Resilience Model, practices such as having good relationships with parents were identified as adequate resilience and adaption among FDY [24,31,35].

### 4.4. School and Community

Many studies have identified factors about both social connections at school and acceptance in the larger community as influencers of resilience. Factors such as school support, school friendships, school connectedness, and sense of belonging at school were operationalized to determine the role of school on resilience [22,24,26,27,32,35]. Furthermore, authors Motti-Stefanidi et al. described educational success and the development of relationships with teachers and school peers as developmental tasks [27].

Similarly, to school factors, community-level factors revolved around inclusion and connection. Factors such as a sense of belonging in the community, intergroup support, community action, social support, high-quality friendships, cumulative exposure to daily stress, quality of peer support, and peer connectedness were found to influence the resilience of FDYs [22,24,26,27,30,31,32].

### 4.5. Exo-Level Factors

Exo-level factors are defined as factors that indirectly influence the individual on a higher level [30]. Unlike micro and meso-level factors, exo-level factors appeared less commonly in the literature review. The exo-level factors described by authors largely related to access to social services and governmental programs that affected FDY. Alarcón et al. and Khan et al. used factors such as immigration programming, housing security, access to education, employment opportunities, and asylum status to describe the effect of the exosystem [24,25,27,36]. Authors Scharpf et al. and Scharpf et al. took a different approach and operationalized exo-level factors as exposure to community violence, availability of parental support networks, and low-support living arrangements [25,30]. Lastly, authors Emuka and Karras described media hypervisibility of black trauma and death and the availability of mental health resources for FDY as exo-level factors [32].

### 4.6. Macro-Level Factors

Identified as factors that impact the broad social, cultural, and economic dynamics of society on a large scale, macro-level factors were the highest-level factors examined from a socio-ecological framework. Several authors connected macro-level factors to acculturative stress, including factors such as exposure to acculturative stressors, language barriers, adaptation to individualistic culture, and adaptation to new social norms and expectations [26,27,28].

Furthermore, the generation of bicultural identity, having a keen sense of national identity, cultural competence in multiple cultures, ability to code switch and integrative acculturation were all identified as desirable outcomes of resilience [23,28,30,32].

### 4.7. Significance of the Results

As highlighted throughout the rapid review, existing models and frameworks of FDY resilience are largely grounded by Bronfenbrenner’s Ecological Systems Theory. Nine out of the twelve included studies included models that are stratified by level of socio-ecological impact, like Bronfenbrenner’s theory [22,24,26,28,30,31,32]. With so many models drawing inspiration from Bronfenbrenner’s initial ideas, two key implications arise.

Firstly, a lack of consistent definitions for concepts such as resilience, resiliency, and various socio-ecological factors made interpreting the operationalization of key factors unclear. Frequently on the micro and meso levels, authors operationalized the same overarching factors using labels that had small, but meaningful, differences in meaning. For example, the micro-level factor self-perception was operationalized as positive self-perception, positive self-esteem, strong sense of identity, and identity affirmation. Even though these concepts were used to assess the same concept, the impact of self-perception on resilience, they all had slight variations in meaning and what was measured.

Secondly, none of the studies identified in this rapid review holistically accounted for cultural influences on FDY resilience building. All twelve of the included studies included frameworks and models that were created by Western scientists. Furthermore, nine of the twelve studies were derived from Bronfenbrenner’s theory, a framework heavily criticized for being not sensitive to cultural contexts [25]. Most of the included studies may share the same limitations due to this fact. Coupled with the implementation of normative principles by researchers like Masten and Suárez-Orozco et al. [35], it is highly unlikely that accurate perspectives of non-Western FDYs are captured.

Our findings notable overlap in the approaches to creating models for resiliency and coping strategies, yet also a lack of distinct specificity. This overlap thus translates into an oversaturation of how researchers are adapting models, yet gaps in that could make implementation challenging. Future work in model adaptation should focus on the implementation of definitions that take into account socio-economic, cultural, and geographical differences. This would remove inconsistencies between authors and would establish a concrete baseline for resilience research. Research should also consider the needs of programmers in governmental and non-governmental organizations who serve different niche populations of youth, such as refugee youth, unaccompanied minors, newcomer youth, and youth who are caretakers for other youth. Research that specifically addresses the contexts of these diverse groups could therefore be utilized to support programmers.

### 4.8. Gaps and Limitations

Although existing models of resilience provide comprehensive overviews of resilience, issues of complexity and lack of causality, cultural generalizability, limited emphasis on the effects of trauma, the linear portrayal of resilience, and dependence on cultural standards of high-income post-migratory settings call attention to the limitations of these models.

Ungar’s ecological perspective of resilience acknowledges the interplay between individual-level factors and multisystemic, community, indirect-external, and socio-cultural factors, but the complexity of the model can be difficult to adapt to practical applications of resilience studies. Furthermore, the model addresses the presence of multisystemic factors but does not provide clear causal relationships between these different variables. The bi-directionality between resilience factors and outcomes poses limitations that can muddle the clarity of interpretation of research findings.

In contrast to Ungar’s model which focuses on providing an overarching, holistic view of resilience, Masten’s model emphasizes the effects of micro-level factors on resiliency building. Masten’s model uses individual characteristics and meso-level factors such as social connectedness, family cohesion, and quality of friendships to determine adequate adaptation in the host country’s society. While these relationships may provide valuable insight into the lower-level interactions that influence resilience, the model disregards the complexity of resilience on higher exo and macro levels. Resilience is influenced by countless factors of varying levels, so having a model that focuses solely on the individual may lose sight of the whole picture of resilience.

Moreover, using developmental checkpoints as indicators of adequate resilience, based on normative standards of host countries suggests that resilience is linear and removes cultural variation among FDY. Firstly, the designation of specific developmental and acculturative tasks may indicate “good” resilience and adaptation among FDY; however, the completion of these tasks does not account for the dynamic nature of resilience or mental health. This generalization may oversimplify the ongoing process of resilience and suggest that adequate resilience can be achieved by following a specific set of linear goals. Secondly, the model revolves around the normative principle and evaluates FDY resilience based on the standards of the host country, irrespective of cultural differences or contexts. This raises challenges in cultural generalizability as most studies evaluated in this review were conducted in high-income destination countries like Germany, Norway, Spain, the United States of America, and Australia (this includes models such as Litman’s COPE inventory, protective, and promotive models of resilience, and Silove’s ADAPT model). These models generally lack cultural responsiveness and operate primarily from a Western perspective, which may not be applicable or respectful to the cultural diversity among FDYs. This gap in critical responsiveness may not align with the lived experiences, cultural beliefs, and practices of FDY from non-Western backgrounds. A Western-centric approach to resilience modelling can lead to the misinterpretation of behaviours and adaptation strategies that are culturally relevant to FDY but do not fit into the Western paradigm of resilience, and vice versa. Suárez et al. acknowledge this gap by highlighting their focus on normative principles [35]. They suggest that developmental tasks should be delegated through communication between parents and teachers and that this collaboration would ideally tailor developmental tasks specific to the cultural context of the individual [35].

Furthermore, the models do not adequately account for the intersectionality of various identity factors such as race, ethnicity, gender, and legal status, which significantly impact FDY’s experiences and resilience. This intersectional approach is necessary to understand the multifaceted nature of resilience among FDY and to develop interventions that are truly inclusive and effective acknowledging the diversity of FDY’s cultural heritage and experiences. Similarly, to Masten’s model of resistance, Schwartz’s reconceptualization of acculturation focuses specifically on individual-level characteristics which may over-simplify the individual experience of acculturation. The four strategies mentioned in the model, assimilation, separation, integration, and marginalization, categorize individuals by distinct acculturation behaviours, but do not account for the complex and dynamic nature of adaptation. Specifically, behaviours assigned to a category (i.e., assimilation) may not linearly correlate with an individual’s acculturation experience. These assumptions suggest that all individuals within the same category experience acculturation events similarly and disregard both higher level socio-cultural factors which may affect FDY of diverse cultural backgrounds differently and ignore individual differences in acculturation. As a result, there is a need for a model that integrates various cultural perspectives, an intersectionality framework, and the dynamic nature of resilience.

Repeated team meetings, cross-checks, and open discussions between the field researchers and the PI ensured the methodological rigour and the risk of bias in the review process. However, given the scope of the review and the timeframe, this additional step was deemed impractical. As a result, the review may include studies with a wide range of methodological strengths and weaknesses, potentially affecting the reliability of the conclusions drawn from this body of literature.

Lastly, the search strategy for the review was limited to the OVID Medline database and open-source peer-reviewed publications. This restriction, while practical, may have inadvertently excluded relevant studies housed in other databases or published in journals not indexed in the searched databases. Such studies could offer different perspectives or pertinent findings to the resilience of FDY, thereby enriching the review’s overall understanding of the topic.

## 5. Conclusions

This review offers a consolidated understanding of existing theoretical frameworks and models of FDY resilience. By identifying successful models and factors, this study lays the foundation for a generalizable model that can guide future interventions aimed at promoting the resilience of FDYs in their host community post-migration. The identified gaps and limitations call for more diverse and culturally inclusive research to ensure the relevance and effectiveness of resilience-building strategies across different contexts.

This review strongly advocates for developing a more generalizable yet culturally inclusive model of resilience, while implementing standardized definitions of resilience and the factors that influence it. This new model should integrate intersectionality, embracing factors like race, ethnicity, gender, and legal status, to reflect the diverse experiences of FDY. The model should move beyond a Western-centric lens and incorporate a variety of cultural perspectives. Incorporating and balancing both micro- and macro-level factors acknowledges the dynamic nature of resilience as an evolving process, thus moving toward a more holistic understanding of FDY resilience. Policymakers and service providers should tailor their services to the specific needs of FDY, who are often assumed to be resilient and can easily cope with adversities.

## Figures and Tables

**Figure 1 ijerph-21-01347-f001:**
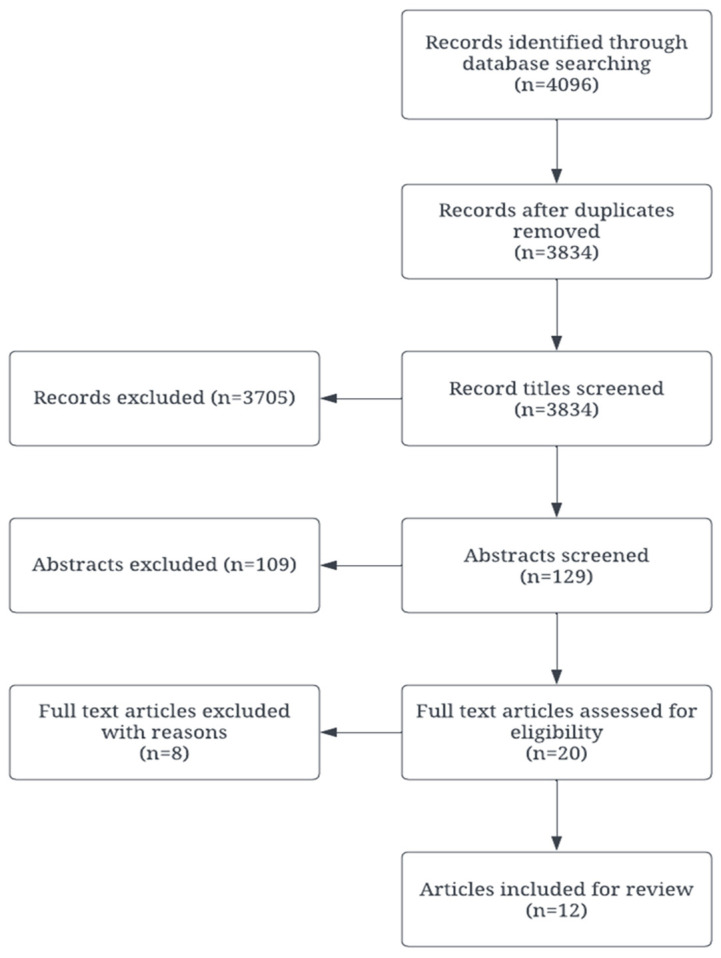
Articles included for the review process.

**Figure 2 ijerph-21-01347-f002:**
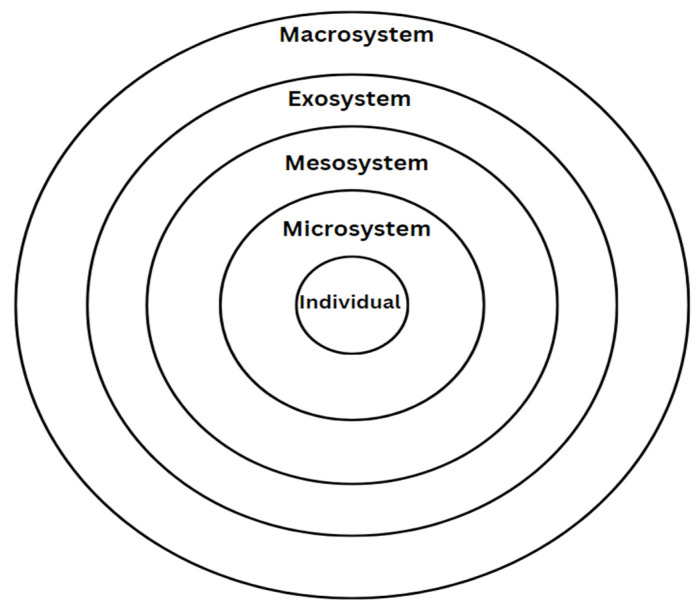
An overview of Bronfenbrenner’s ecological systems theory [34].

## Data Availability

No new data were created or analyzed in this study. Data sharing is not applicable to this article.

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
