# Peer review of "Resilience Mechanisms and Coping Strategies for Forcibly Displaced Youth: An Exploratory Rapid Review"

_ijerph, 2024, doi:10.3390/ijerph21101347_

Round 1

Reviewer 1 Report

Comments and Suggestions for Authors

The manuscript “Resilience Mechanisms and Coping Strategies for Forcibly Displaced Youth Experiencing Social Isolation: An Exploratory Rapid Review” is a well-written review article dealing with an important and contemporary subject of resilience mechanisms of forcibly displaced young people – migrant category which is on the rise in recent years. Even though the article brings the results of a rapid review of twelve published articles, it sets the ground for further research and the possible introduction to a systematic review of the subject. Also, the added value of the manuscript is a clear and detailed limitation section which precisely defines shortcomings of the research and sets the framework for their overcoming in future research.

The manuscript is clearly written and corresponds to the usual structure of scientific publishing. There are two minor issues the authors should consider before the paper is published.

Firstly, in the Methods section, I suggest the authors add a bit more information on the thematic analysis, to describe how the analysis was conducted, which were the major themes (thematic clusters), whether there was any coding procedure etc.

In the Results section two of the analysed models lacked a direct connection to the reviewed articles: Litman’s COPE Inventory and Silove’s ADAPT Model. Some references to the reviewed articles should be introduced into these subsections.

Author Response

Reviewer 1

“Firstly, in the Methods section, I suggest the authors add a bit more information on the thematic analysis, to describe how the analysis was conducted, which were the major themes (thematic clusters), whether there was any coding procedure etc.”

Thank you for this feedback. To ensure the viewer has a full understanding of our methodology, we have added additional sentences to the Methods section, stating: "The review compiled emerging themes, which were carefully categorized and analyzed based on factors contributing to resilience. These were further classified into thematic clusters that emerged from the analysis.”

In the Results section two of the analysed models lacked a direct connection to the reviewed articles: Litman’s COPE Inventory and Silove’s ADAPT Model. Some references to the reviewed articles should be introduced into these subsections.”

We appreciate this suggestion. The subcategorizations of Litman’s COPE Inventory and Silove’s ADAPT Model emerged directly from the articles reviewed. We have referenced the articles that incorporate these models in the “Current Models of Resilience” sub-section. After discussing with the team, we feel that this initial reference is sufficient in linking the review articles and their respective models.

Reviewer 2 Report

Comments and Suggestions for Authors

Some more relevant references would improve the quality of the article. Have pointed out in the pdf. Also, the authors can explain about the geographic region as mentioned in this paper? why not asia and Europe has been given focus? 

Author Response

Reviewer 2

“Some more relevant references would improve the quality of the article. Have pointed out in the pdf”

Thank your for providing references Ullah & Chattoraj (2022) and Chattoraj (2024). We have Included these references in the two suggestions within the introduction. Our team did not find it appropriate to reference Ullah & Chattoraj (2022) in the results section, as this article did not emerge from our search.

“[106-107] This had been mentioned earlier too”

We appreciate this suggestion. Following your insightful comment, we have removed our initial definition of youth.

Also, the authors can explain about the geographic region as mentioned in this paper? why not asia and Europe has been given focus?”

Thank you for this comment/question regarding discussed regions. The articles included in our analyses were the result of our search parameters and the inclusion criteria. Therefore, specific regions were not deliberately excluded. Future research could suggest incorporating and modifying keywords or approaches to ensure broader geographic representation. Furthermore, lines 140-160 aim to demonstrate the diversity within the reviewed articles, which include mention of Asian and European settings.

Reviewer 3 Report

Comments and Suggestions for Authors
  • General comments thank you for the opportunity to review this important and timely article.
    Article: This is a very important rapid review on "Resilience Mechanisms and Coping Strategies for Forcibly Dis placed Youth Experiencing Social Isolation: An Exploratory  Rapid Reviewhigh" and adds in depth understanding about theoretical frameworks on youth resilience factors. 
    Review: The review uses PRISMA guidelines for reporting. Justification for rapid review is made related to informing policy and practice. Key search terms narrow the focus of the review with a clear definition of youth. Inclusion and exclusion criteria is justified. Research question guiding the review is included "What are  the current models of resilience regarding refugee youth? What are the factors included  in each model? What are the gaps and limitations in each model?.
  • Given the study title it is unclear how the focus on social isolation was included in the findings and/or analysis? . The authors may consider a sentence or two related to this in the findings which describe the models.
  • Specific comments:
  • The construct of belonging is mentioned on line 279 under findings. It may bring more integration to the findings to understand how social isolation factors are included or not across the 9  models identified that met the inclusion criteria. The review is overall relevant but more clarity about how policy and practice could implement the findings in a more pragmatic way. For example in settlment service sectors working with youth who have experienced forced displacement. 
  • Overall, the manuscript is clear and well written. The concept of intersectionality comes later in the manuscript around lines 528- the authors could consider moving this theoretical lens to the background as well. For example, defining the term and it's origins and if that was a lens used to analyze the models and papers, it is clear that it was because that is one of the recommendations. The point is that readers may appreciate a definition of intersectionality or how it was applied in the review.
  • The authors recommend that future work could include standard definitions to establish baseline of resilience, this critique may require further clarification because a one size fits all approach may not be congruent with an intersectionality framework? -later (lines 509-510)generalization of resilience is described as potentially oversimplifying process of resilience; is there a contradiction or tension here that can be further clarified?
  • Finally, the point about western centric models is well recieved, might there be a role for understanding the mechanisms that promote or support social inclusion? 
  • These recommendations are for consideration and may improve overall clarity of the results. And can be considered for minor revisions.

Thank you for this comprehensive work and contribution to understanding forced displaced youth resiliency factors.

Author Response

Reviewer 3

Given the study title it is unclear how the focus on social isolation was included in the findings and/or analysis? . The authors may consider a sentence or two related to this in the findings which describe the models.”

Thank you for your feedback on our title. This was extremely helpful in considering the objective our paper.  As none of the reviewed articles explicitly discussed social isolation, we cannot substantiate its inclusion in the findings. After discussing with our team we will remove the mention of of "experiencing social isolation," in our title. We sincerely appreciate the feedback.

The construct of belonging is mentioned on line 279 under findings. It may bring more integration to the findings to understand how social isolation factors are included or not across the 9  models identified that met the inclusion criteria.”

Thank you for taking the time to provide feedback in our findings. Since we have decided to remove social isolation as a framing for our paper, we will not be including a discussion on social isolation factors.

“The review is overall relevant but more clarity about how policy and practice could implement the findings in a more pragmatic way. For example in settlment service sectors working with youth who have experienced forced displacement. “

Thank you for bringing up this point about applying our findings into practical settings. The last paragraph in the conclusion is a policy-oriented statement which lays out the several factors which should be considered in policy-making, service provision and other practical settings. We added this sentence to the last paragraph in the conclusion “Policymakers and service providers should tailor their services to the specific needs of FDY, who are often assumed to be resilient and can easily cope with adversities.”

“The concept of intersectionality comes later in the manuscript around lines 528- the authors could consider moving this theoraetical lens to the background as well. For example, defining the term and it's origins and if that was a lens used to analyze the models and papers, it is clear that it was because that is one of the recommendations. The point is that readers may appreciate a definition of intersectionality or how it was applied in the review.”

We appreciate your suggestion in clarifying intersectionality and how it was utilized in our paper. As we conducted a rapid review, we were not aiming to apply intersectionality in our analysis or interpretation. We had simply brought up intersectionality to demonstrate the intersecting factors involved.

The authors recommend that future work could include standard definitions to establish baseline of resilience, this critique may require further clarification because a one size fits all approach may not be congruent with an intersectionality framework? -later (lines 509-510)generalization of resilience is described as potentially oversimplifying process of resilience; is there a contradiction or tension here that can be further clarified?”

We appreciate your feedback on clarifying the standardized definitions reccomendation. We agree that standard definitions of resilience should consider contextual factors, and should definitely not follow a “one-size-fits-all” approach. To avoid the appearance of contradiction and clarify our stance, we will clarify that while resilience research benefits from a baseline definition, this should be flexible and adaptable to diverse cultural and contextual realities. We have extended the relevant sentence by adding “Future work in model adaptation should focus on the implementation of standardized definitions that take into account socio-economic, cultural and geographical differences.”

“Finally, the point about western centric models is well recieved, might there be a role for understanding the mechanisms that promote or support social inclusion?”

Thank you for including this suggestion. Given that the nature of our search and the articles themselves did not speak on how to promote social inclusion, our team feels that it is not appropriate to comment on mechanisms that can promote social inclusion.

Round 2

Reviewer 2 Report

Comments and Suggestions for Authors

Accepted